# Human responses to the DNA prime/chimpanzee adenovirus (ChAd63) boost vaccine identify CSP, AMA1 and TRAP MHC Class I-restricted epitopes

Harini Ganeshan[1,2☯], Jun Huang[1,2☯], Maria Belmonte[1,2], Arnel Belmonte[1,3], Sandra Inoue[1,3], Rachel Velasco[1,3], Santina Maiolatesi[1,2], Keith Limbach[1,2], Noelle Patterson[1,2], Marvin J. Sklar[1], Lorraine Soisson[4], Judith E. Epstein[5], Kimberly A. Edgel[1], Bjoern Peters[6], Michael R. Hollingdale[1,2], Eileen Villasante[1], Christopher A. Duplessis[1], Martha Sedegah[1] *

1 Naval Medical Research Command, Silver Spring, Maryland, United States of America, 2 The Henry M. Jackson Foundation for the Advancement of Military Medicine, Inc., Bethesda, Maryland, United States of America, 3 General Dynamics Information Technology, Falls Church, Virginia, United States of America, 4 United States Agency for International Development (USAID), Washington, DC, United States of America, 5 Laboratory of Malaria Immunology and Vaccinology, National Institute of Allergy and Infectious Diseases, National Institutes of Health, Bethesda, Maryland, United States of America, 6 La Jolla Institute of Allergy and Immunology, La Jolla, California, United States of America

☯ These authors contributed equally to this work.
* martha.sedegah.civ@health.mil

**Data Availability Statement:** All relevant data are within the manuscript and its Supporting Information files.

## Abstract

### Background

A three-antigen DNA-prime/chimpanzee adenovirus 63 (ChAd63) boost vaccine containing pre-erythrocytic *Plasmodium falciparum* (Pf) circumsporozoite protein (CSP), Pf apical membrane antigen-1 (AMA1) and malaria multiple epitopes (ME) fused to Pf thrombospondin-related adhesion protein (ME-TRAP) elicited higher vaccine efficacy (VE) in an open label, randomized Phase 1 trial against controlled human malaria infection (CHMI) than the two-antigen vaccine DNA/Human Adenovirus 5 (HuAd5) containing CSP and AMA1. The objective of this follow-up study was to determine whether responses to CSP, AMA1 or TRAP MHC Class I-restricted epitopes were associated with VE.

### Methodology

Protected (n = 6) and non-protected participants (n = 26) were screened in FluoroSpot interferon gamma (IFN-γ) and Granzyme B (GzB) assays using antigen-specific 15mer peptide subpools spanning CSP (n = 9 subpools), AMA1 (n = 12 subpools), and TRAP (n = 11 subpools). Individual antigen-specific 15mers in the subpools with strong responses were then deconvoluted, evaluated for activities, and MHC Class I-restricted epitopes within the active 15mers were predicted using NetMHCpan algorithms. The predicted epitopes were synthesized and evaluated in the FluoroSpot IFN-γ and GzB assays.

**Funding:** This study was funded by the United States Agency for International Development (USAID) Malaria Vaccine Development Program through Interagency Agreement (IAA) AID-GH-T-17-00002 between USAID and the NMRC. LS participated in the concept development of this project, but otherwise had no role in study design, data collection and analysis, decision to publish, or preparation of the manuscript.

**Competing interests:** The authors have declared that no competing interests exist.

## Results

Protected and some non-protected participants had similar responses to individual antigen-specific peptide subpools, which did not distinguish only protected participants. However, deconvoluted antigen-specific positive subpools with high magnitudes of responses revealed individual 15mer peptides containing specific and/or predicted MHC Class I (HLA) epitopes. Responses to epitopes were either IFN-γ-only, IFN-γ and GzB, or GzB-only. Due to limitation of cells, most of the analysis concentrated on the identification of protection associated AMA1 epitopes, since most of the predominant pool specific responses were generated against AMA1 15mer subpools. Furthermore, we previously identified protection associated HLA class I-restricted epitopes in a previous gene-based vaccine trial. Seven predicted minimal epitopes in AMA1 were synthesized and upon testing, five recalled responses from protected participants confirming their possible contribution and association with protection, and two recalled responses from non-protected participants. Two protection-associated epitopes were promiscuous and may have also contributed to protection by recognition of different HLA alleles. In addition, strongly positive antigen-specific 15mers identified within active antigen-specific subpools contained 39 predicted but not tested epitopes were identified in CSP, AMA1 and TRAP. Finally, some non-protected individuals recognized HLA-matched protection-associated minimal epitopes and we discuss possible reasons. Other factors such as HLA allele fine specificity or interaction between other HLA alleles in same individual may also influence protective efficacy.

## Conclusions

This integrated approach using immunoassays and bioinformatics identified and confirmed AMA1-MHC Class I-restricted epitopes and a list of predicted additional epitopes which could be evaluated in future studies to assess possible association with protection against CHMI in the Phase 1 trial participants. The results suggest that identification of protection-associated epitopes within malaria antigens is feasible and can help design potent next generation multi-antigen, multi-epitope malaria vaccines for a genetically diverse population and to develop robust assays to measure protective cellular immunity against pre-erythrocytic stages of malaria. This approach can be used to develop vaccines for other novel emerging infectious disease pathogens.

## Introduction

Malaria remains a significant threat world-wide, especially to children, and a vaccine is essential to control this disease. Malaria is transmitted by bites of infected *Anopheles* mosquitoes that inject sporozoites that rapidly invade liver cells, asexually develop and release merozoites that cause repeated cycles or red blood cell infection associated with clinical disease. A vaccine that prevented red cell infection would prevent clinical malaria that is particularly serious in young children. The World Health Organization has approved two malaria vaccines for use in certain regions. Both vaccines (RTS,S/AS01, and R21/Matrix-M) are made up of a hepatitis B surface antigen and a *Plasmodium falciparum* (Pf) sporozoite antigen (circumsporozoite protein [CSP]). Both vaccines reduce severe malaria cases by 75% when given seasonally in areas of highly seasonal transmission (WHO Malaria Vaccines RTS,S and R21), by anti-CSP

antibodies. However, a completely effective malaria vaccine is still not available, and other potential vaccines are being developed (such as a live, attenuated whole sporozoite vaccine [1]) that target sporozoites and liver stages.

Our approach is aimed at developing a multiantigen-multi epitope vaccine that would elicit anti-sporozoite antibodies and/or cellular responses that target infected liver cells, preventing red cell infection. Here we focus on better understanding of cellular responses to vaccine antigens. CD8+ T lymphocytes are critical mediators of long-term protective immunity against malaria [2,3], killing intracellular liver stage parasites through interferon-gamma (IFN-γ), and/or other cytokines, and immune mediators, such as Granzyme B (GzB) that require direct contact with infected hepatocytes [3–5]. These responses were successfully elicited using gene-based heterologous prime-boost approaches that have induced vaccine efficacy (VE) against controlled human malaria infection (CHMI) [2,6–9]. Previously, we used a recombinant DNA plasmid-prime/recombinant human adenovirus serotype 5 (HuAd5) boost vaccine, encoding two pre-erythrocytic antigens, CSP and Pf apical membrane antigen-1 (AMA1), that sterilely protected 4/15 (27%) participants against CHMI [10]. However, concerns about the safety of HuAd5 [11], and effects of naturally-acquired neutralizing antibodies on HuAd5 vaccine immunogenicity [12], led to the development of an alternative adenovirus vector, chimpanzee adenovirus 63 (ChAd63), at the University of Oxford [13,14]. They showed that a ChAd63 prime/modified vaccinia virus Ankara (MVA) boost vaccine expressing the PfME-TRAP antigen (a string of 20 malarial T- and B- cell epitopes, ME, fused to the Pf thrombospondin-related adhesion protein (TRAP) elicited monofunctional CD8+ IFN-γ T cell responses that correlated with sterile protection achieved in 3/14 (21%) participants [15]. In collaboration with the University of Oxford, we replaced HuAd5 with ChAd63 and repeated our study using the DNA/ChAd63 two antigen (CSP and AMA1 = CA) vaccine and a novel three antigen (CSP, AMA1, and ME-TRAP = CAT) vaccine [16]. After CHMI, 1/16 (6%) immunized participants in the CA group, and 5/16 (31%) participants in the CAT group did not develop parasitemia. We concluded that adding ME-TRAP to a two-antigen (CA) to produce a three-antigen CAT formulation increased VE [16].

In the DNA/HuAd5 vaccine trial, antibody responses were not correlated with protection [10], so we focused on cellular responses in the current trial. Participants' peripheral blood mononuclear cells (PBMCs) taken pre-CHMI were analyzed for IFN-γ responses using ELI-Spot IFN-γ assays and peptide subpools containing 15mers spanning each antigen. Three of four protected participants had higher IFN-γ responses to CSP and AMA1-peptide subpools than non-protected participants, whereas one protected participant had low activities to both antigens [10]. An integrated approach using MHC Class I prediction algorithms [17] and ELI-Spot IFN-γ assays suggested that these protective responses were directed against CSP and/or AMA1 MHC Class I-restricted epitopes [18]. Synthesized AMA1 MHC Class I epitopes restricted by HLA A03 or B58 supertypes (ST) recalled IFN-γ robust responses in three of four protected participants' PBMC [18]. We hypothesized that the DNA/HuAd5 vaccine's sterile protection to malaria was mediated by monofunctional effector CD8+ T cells that target specific MHC Class I-restricted epitopes in the vaccine antigens.

We have further explored these studies using DNA/ChAd63 CA and CAT vaccine trial samples taken pre-CHMI to determine whether the previously identified or novel MHC Class I-restricted epitopes are associated with VE. We have reported that like the DNA/HuAd5 trial, VE in the DNA/ChAd63 trial was not correlated with antibody responses [16]. Our hypothesis is that protected participants develop a predominant antigen-specific response to an individual 15mer peptide subpool containing MHC Class I-restricted epitope(s) that match the participant's HLA alleles. To test the hypothesis, we predicted the binding affinities of CSP, AMA1, and TRAP epitopes within 15mer peptide components of strongly active subpools using the

NetMHCpan algorithm [17]. Confirmation of epitope specificity used FluoroSpot IFN-γ and GzB assays with individual 15mer peptides within deconvoluted dominant antigen-specific peptide subpools and synthesized predicted minimal epitopes. Due to cell limitation, epitope confirmation was done for AMA1 while we identified immunodominant regions of CSP and TRAP vaccine antigens in the form of strongly active 15mer peptides containing predicted minimal epitopes. We also discuss whether other factors such as HLA allele specificity and interaction between multiple HLA-restricted epitope responses in same individual affect protection.

The antibodies produced by this vaccine is too low to play a significant role in sterile protection in this study. Furthermore, due to limitation of cells, we were unable to analyze and confirm additional cellular responses against predicted epitopes within the other vaccine antigens, CSP and TRAP, and we could not determine the contribution these other antigen specific cellular responses made towards protection.

## Methods

### Objectives

The objectives were to evaluate antigen-specific FluoroSpot IFN-γ and GzB activities induced by the DNA/ChAd63 CA and CAT vaccines in protected and non-protected participants using pre-CHMI samples [16] and to identify and confirm predicted MHC Class I-restricted epitopes. Not all subjects could be evaluated due to limitation of cells. The DNA vaccine constructs used the 3D7 strain of Pf CSP, AMA1 or TRAP, and the ChAd63 vaccine constructs used the 3D7 strain of CSP and AMA1 and T9/96 strain of TRAP [16]. TRAP amino acid sequences from T9/96 and 3D7 differ by 6.1% [19]. CHMI was conducted using the Pf 3D7 strain [16].

### Human ethics statement

The clinical trial was conducted at the Naval Medical Research Center (NMRC), now Naval Medical Research Command, Clinical Trials Center from 2018–2019 (now Translational Clinical Research, TraCR); CHMIs were conducted at the Walter Reed Army Institute of Research (WRAIR) secure insectary. The study protocol was reviewed and approved by the NMRC Institutional Review Board (IRB) in compliance with all applicable federal regulations governing the protection of human research participants. WRAIR and NMRC each hold a Federal-wide Assurance from the Office of Human Research Protections (OHRP) under the Department of Health and Human Services. NMRC also holds a Department of Defense/Department of the Navy Assurance for human participant protections. All key personnel were certified as having completed mandatory human participants' protection curricula and training under the direction of the WRAIR IRB and Human Subjects Protections Branch (HSPB) or the NMRC IRB and Office of Research Administration (ORA). All potential study participants provided written, informed consent before screening and enrollment and had to pass an assessment of understanding. The study was conducted according to the Declaration of Helsinki, as well as principles of Good Clinical Practices under the United States Food and Drug Administration (FDA) Investigational New Drug (IND) application IND 17572. The trial was performed under an IND allowance by the FDA and was registered on ClinicalTrials.gov (NCT03341754).

### Human participants

The full clinical details of this trial, including patient recruitment and flow, safety and tolerability have been previously reported [16]. For this study 32 participants were available: 16 were

immunized with DNA/ChAd63 CA (one was fully protected against CHMI), and 16 were immunized with DNA/ChAd63 CAT (five were fully protected against CHMI). None of the non-protected participants showed a significant delay to parasitemia as previously defined [20].

## HLA typing of participants

The HLA A* and HLA B* alleles were analyzed by high resolution typing at the Georgetown University Marrow Donor Service and Testing Center and are shown in S1 Table.

## Immunological samples

Peripheral blood mononuclear cells (PBMCs) were collected pre-immunization and at 27 days post ChAd63 immunization / five or six days pre-CHMI. [16]. We used freshly isolated PBMC for assays with antigen-specific peptide subpools to identify immunodominant regions of vaccine antigens and cryopreserved and thawed PBMCs for assays with 15mer peptides and synthesized minimal epitopes within the identified dominant subpools. We have previously shown that ELISpot activities of fresh and cryopreserved PBMC may differ in magnitude but not antigen specificity [18,21]. Since fresh PBMC were used for initial evaluations with subpools, and cryopreserved PBMC were used for all 15mer and epitope evaluations, we consider that both evaluations are comparable.

## CSP, AMA1 and TRAP peptide subpools

15mer peptides were synthesized by Mimotopes, VIC, Australia. Minimal epitopes (9-10mer) were synthesized by Alpha Diagnostics Intl Inc, San Antonio, TX, USA and Mimotopes, VIC, Australia. Full length CSP and AMA1 were covered by a series of 65 (CSP) and 153 (AMA1) 15mer amino acid (aa) sequences overlapping by 11 aa. CSP 15mers were combined into 9 individual peptide subpools (Cp1-Cp9), and AMA1 15mers were combined into 12 individual peptide subpools (Ap1-Ap12) [7]. Full length TRAP strain T9/96 (TT) and TRAP strain 3D7 (TD) were covered by 20mer peptides overlapping by 10 aa; 56 TRAP TT and 50 TRAP TD peptides were combined into 6 (TT1-TT6) or 5 (TD1-TD5) individual peptide subpools, as TD6 contained the same aa as TT6 [22]. In deconvolution experiments, 15mer peptides in dominant CSP and AMA1 subpool were evaluated individually. However, for deconvolution of TRAP TD subpools, an equivalent series of 15mer peptides (PfSSP2, designated as SS) overlapping by 11 aa were used rather than the 20mer peptides. There were no comparable series of 15mers for TRAP TT; however, T9/96 and 3D7 TRAP sequences share 94% amino acids [19].

## IFN-γ and GzB FluoroSpot assay

Fresh or cryopreserved [16] PBMCs were tested for IFN-γ and/or GzB secretion using FluoroSpot assays as previously described [23]. Briefly, PBMCs were incubated with stimulants for 40–42 h at 37°C in 5% $CO_2$ on capture plates coated with both anti-IFN-γ and anti-GzB antibodies, and activities expressed as spot forming cells (sfc) secreting IFN-γ and/or GzB per $10^6$ PBMC. A positive response to each stimulant was defined with background subtracted as previously described [10]. A participant was considered positive to a particular antigen if there was a positive response to at least one peptide subpool for that antigen. An immunodominant response of a positive antigen-specific peptide subpool was defined as representing 50% or more of the sum of total responses to all subpools of the specific antigen.

## Identification of minimal antigen specific MHC Class I restricted epitopes within immunodominant peptide subpools of CSP, AMA1, and TRAP

MHC Class I-restricted minimal epitopes from positive immunodominant antigen-specific peptide subpools were predicted using the NetMHCpan-4.0 as we have previously described [18,21]. Immunodominant antigen-specific subpools were deconvoluted and individual 15mer peptide components were evaluated in FluoroSpot assays. Predicted epitopes within the active AMA1 15mers were synthesized and evaluated in FluoroSpot assays [23].

## Results

### Summed antigen-specific 15mer subpool FluoroSpot IFN-γ and GzB responses to CSP, AMA1 and TRAP

Using fresh PBMCs, summed IFN-γ and GzB responses to antigen-specific peptide subpools are shown in Fig 1. We define summed vaccine-antigen response for each participant as the sum of responses against subpools for each antigen, represented as color-coded stacked bars for each participant (Fig 1). Positive IFN-γ responses predominated with or without GzB responses. GzB responses only occurred in participants with positive IFN-γ responses, except for two participants in the CA cohort with negative IFN-γ responses but low positive GzB responses to CSP.

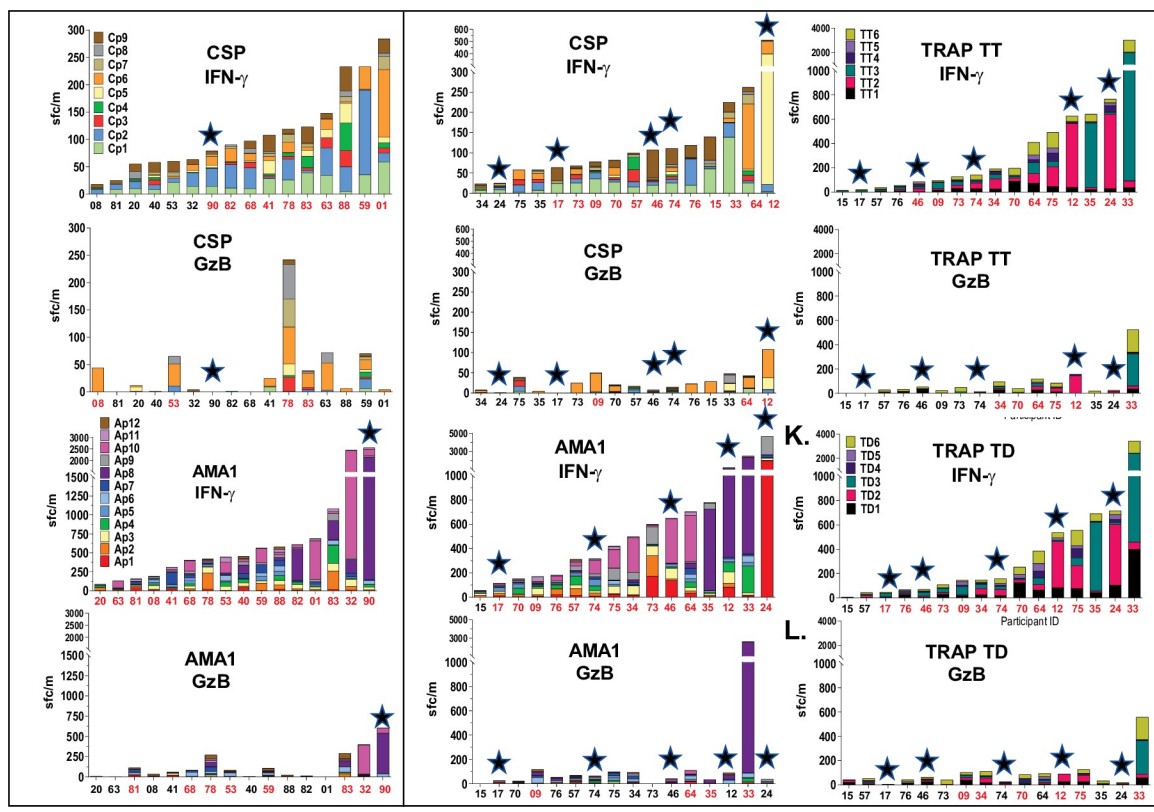

**Fig 1. CA and CAT cohorts: Post-ChAd63 boost/pre-CHMI FluoroSpot IFN-γ and GzB activities to CSP, AMA1 and TRAP subpools.** FluoroSpot IFN-γ and GzB activities to subpools are shown as color-coded stacked bars and arranged in ascending order of summed total IFN-γ and GzB responses for each vaccine antigen. ★ **Protected subjects.** CA (Panels A—D); CAT (Panels E—L). X-axis represents participant identification numbers and participants with positive summed total response to an antigen in red font and not positive summed total responses in black font. We define response to an antigen as positive if a positive response (based on our pre-determined criteria) is induced against one or more of the antigen-specific subpools tested.

**Table 1. Cohort CA: FluoroSpot IFN-γ and GzB responses of protected participant v90 (HLA A\*33:03 (A03)/A\*33:0 (A03), B\*44:03 (B44)/B\*58:01 (B58) to AMA1 Ap8, Ap10 and Ap11 peptide subpools, 15mer peptides, and tested predicted minimal epitopes.**

| Pool/ 15mer | 15mer Sequence | IFN-γ sfc/m | GzB sfc/m | HLA restriction | 15mer Peptide | Epitope | IFN-γ sfc/m | GzB sfc/m |
|---|---|---|---|---|---|---|---|---|
| **Ap8** | | **418** | **175** | | | | | |
| A97 | FKADRY(**KSHGKGYNW**) | **585** | **95** | **B\*58:01** | **A97** | **KSHGKGYNW** | **467** | 87 |
| A98 | (**RYKSHGKGYNW**)GNYN | **110** | 5 | **B\*58:01** | **A98** | **YKSHGKGYNW** | **263** | 87 |
| | | | | | A98 | RYKSHGKGY | 10 | 20 |
| A103 | (**CEIFNVKPT**)CLINNS | 3 | **65** | **B\*45:01** | A103 | CEIFNVKPT | Not tested | Not tested |
| **Ap10** | | 0 | 0 | | | | | |
| A126 | TC(**RFFVCKCVER**)RAE | 48 | 0 | **A\*33:03** | **A126** | **RFFVCKCVER** | **180** | **123** |
| **Ap11** | | 0 | 0 | | | | | |
| A136 | TYDKMK(**IIIASSAAV**) | 15 | 0 | **A\*02:01** | A136 | IIIASSAAV | 7 | 0 |
| A137 | MK(**IIIASSAAVAV**)LA | 0 | 3 | **B\*58:01** | | | | |

PBMCs were collected from participant v90 post-ChAd63/pre-CHMI. All 15mer peptides within Ap8, Ap10, and Ap11 were evaluated using FluoroSpot assays. Positive activities against subpools and/or 15mers are shown in bold. Predicted epitopes within positive 15mers are shown in bold within parentheses and underlined. Five predicted epitopes were synthesized and evaluated using the FluoroSpot assay.

<u>CA cohort</u>: the single protected participant (v90) and one non-protected participant (v32) had the highest summed IFN-γ and GzB responses to AMA1 (Fig 1C and 1D). <u>CAT cohort</u>: two of five protected participants (v12, v24) had the highest summed IFN-γ and/or GzB responses to CSP (Fig 1E and 1F) and among the top three for AMA1; Fig 1G). However, a non-protected CAT participant (v33) had the highest summed GzB responses to AMA1 (Fig 1H), and the highest summed IFN-γ and summed GzB responses to 3D7 TRAP (Fig 1K and 1L).

Therefore, summed responses to antigen subpools did not fully distinguish between protected and non-protected participants unlike summed responses in the DNA/HuAd5 trial [10].

We selected protected and non-protected participants with the highest FluoroSpot activities in Fig 1, for whom we had sufficient cryopreserved PBMCs, to evaluate 15mers in highly active deconvoluted peptide subpool. Cryopreserved PBMCs were limited, and we recognize that these outcomes are preliminary and would require confirmation in any follow-up vaccine studies. Predicted MHC Class I-restricted epitopes within strongly active 15mer peptides were synthesized and evaluated for IFN-γ and GzB responses in matching participants.

The main <u>Tables</u> in the manuscript show IFN-γ and GzB responses to antigen-specific peptide subpools and only positive responses to deconvoluted 15mers within that subpool, as well as responses to tested minimal epitopes within the active 15mer are shown. The <u>Supplementary Tables</u> show the full range of positive and negative responses to the individual 15mers components of positive subpools for the three vaccine antigens that include CSP, AMA1, and TRAP. We have also bolded the predicted epitopes within the active 15mers.

## CA Cohort: Identification of MHC Class I-restricted epitopes

### Protected CA participant v90 (HLA A\*33:03 (A03)/A\*33:03 (A03), HLA B\*44:03 (B44)/ B\*58:01 (B58): (Tables 1 and S2)

*Synthesized and confirmed predicted minimal epitope within 15mer component of AMA1 subpool.* **AMA1 Ap8 subpool**: IFN-γ and GzB responses to Ap8 15mer peptides A97 and A98 that contained predicted HLA B\*58:01 (B58 ST)-restricted epitope **KSHGKGYNW** that

**Table 2. Cohort CA: FluoroSpot IFN-γ and GzB responses for non-protected participant v32 (HLA A*03:03 (A03)/A*03:03) (A03), B*07:02 (B07)/B*07:02 (B07) to AMA1 Ap8 and Ap10 peptide subpools, 15mer peptides, and predicted minimal epitopes.**

| Pool/ 15mer | 15mer Sequence | IFN-γ sfc/m | GzB Sfc/m | HLA restriction | 15mer Peptide | Epitope | IFN-γ sfc/m | GzB Sfc/m |
|---|---|---|---|---|---|---|---|---|
| **Ap8** | | 35 | 0 | | | | | |
| A95 | (**SAFLPTGAFK**)ADRYK | **65** | 0 | **A*03:01** | A95 | **SAFLPTGAFK** | **105** | 0 |
| **Ap10** | | 735 | 48 | | | | | |
| A125 | VSN(**STCRFFVCK**)CVE | **580** | 8 | **A*03:01** | A125 | **STCRFFVCK** | **1418** | **405** |
| | VS(**NSTCRFFVCK**)CVE | | | **A*0301** | | **NSTCRFFVCK** | **718** | **113** |
| A126 | TC(**RFFVCKCVER**)RAE | **108** | 3 | **A*03:01** | A126 | RFFVCKCVER | 13 | 0 |

PBMCs were collected from participant v32 post-ChAd63/pre-CHMI. All 15mer peptides within AMA1 Ap8 and Ap10 subpools were evaluated in FluoroSpot assays. Positive activities are shown in bold. Predicted epitopes within positive 15mers are shown in bold with parenthesis and underlined. Each predicted epitope shown was synthesized and evaluated in FluoroSpot assays.

recalled IFN-γ but not GzB responses from v90. **KSHGKGYNW** was also associated with protection of an HLA B58 participant in the DNA/HuAd5 trial [10]. Interestingly, 15mer peptide A98 also contained another predicted HLA B*58:01 (B58 ST)-restricted epitope **RYKSHGKGY,** which did not recall responses in the same HLA matched v90. A third 15mer peptide within Ap8 subpool, A103, also induced positive GzB responses but a predicted HLA B*45:01 (B44 ST)-restricted epitope, **CEIFNVKPT** which also matched v90's HLA ST but was not tested (Tables 1 and S2).

*AMA1 Ap10 subpool*: IFN-γ responses to Ap10 subpool 15mer peptide A126 that contains predicted HLA A*33:03 (A03 ST)-restricted epitope **RFFVCKCVER** that recalled IFN-γ and GzB responses.

*Synthesized and not confirmed predicted minimal epitope within 15mer component of AMA1 subpool*. **AMA1 Ap11 subpool**: Although IFN-γ responses to Ap11 subpool, and two overlapping 15mer peptides A136 and A137 within Ap11 were negative, both 15mers contain epitope **IIIASSAAV,** predicted to bind both HLA A*02:01 (A02 ST) and B*58:01 (B58 ST). Since v90 expresses B58, the epitope was tested but did not recall responses (Tables 1 and S2).

**Non-protected participant v32 (HLA A*03:03 (A03)/A*03:03) (A03), B*07:02 (B07)/ B*07:02 (B07): (Tables 2 and S3)**

*Synthesized and confirmed predicted minimal epitope within 15mer component of AMA1 subpool*. **AMA1 Ap8 subpool**: IFN-γ responses to Ap8 15mer peptide A95 that contains predicted HLA A*03:01 (A03 ST)-restricted epitope **SAFLPTGAFK** that recalled IFN-γ responses.

**AMA1 Ap10 subpool**: IFN-γ responses were made to Ap10 15mer peptide A125 that contains predicted HLA A*03:01 (A03 ST)-restricted epitopes **STCRFFVCK** and (**N**) **STCRFFVCK,** that recalled IFN-γ and GzB responses. A third predicted HLA A*03:01 (A03 ST)-restricted epitope RFFVCKCVE, within 15mer peptide A126, was not active. This shows that within the same individual, reactivities against predicted epitopes with the same restriction vary and some dominate over others.

## Interpretation of responses of CA Cohort protected v90 and non-protected v32

- Protection of v90 is associated with: IFN-γ and GzB responses to the HLA B58-restricted epitope KSHGKGYNW (within AMA1 subpool Ap8) and IFN-γ and GzB responses to the HLA A03-restricted epitope RFFVCKCVER (within AMA1 subpool Ap10).

- The fine specificity of HLA alleles may influence whether responses are associated with protection:

  ○ Non-protected v32 in this trial and a protected participant in the DNA/HuAd5 trial recognized the HLA A03 supertype-restricted epitope **STCRFFVCK** (within AMA1 subpool Ap10. However, v32 expressed HLA A*03:01 allele whereas the DNA/HuAd5 participant expressed HLA A*11:01 allele, both members of the HLA A03 ST [10], suggesting the fine specificity of HLA alleles may be important for protection.

  ○ Protected v90 recognized the HLA A03 restricted epitope **RFFVCKCVER** (within AMA1 subpool Ap10), but non-protected v32 did not recognize the HLA A03 restricted epitope **RFFVCKCVER**. Since v90 expresses HLA A*33:03 and v32 expresses HLA A*03:01, both members of the HLA A03 ST, it appears that recognition of the same epitope by the different HLA alleles, HLA A*33:03 but not HLA A*03:01-restricted responses to **RFFVCKCVER** could affect association with protection.

- Magnitude of IFN-γ responses did not influence association with protection: HLA A*03:01 (A03 ST)-restricted epitope **SAFLPTGAFK** (within AMA1 subpool Ap8) recalled lower IFN-γ responses whereas HLA 03:01-restrcted epitope **STCRFFVCK** recalled higher magnitude of IFN-γ and GzB responses from non-protected v32. However, the small sample size of tested epitopes and number of study participants with adequate PBMC precludes a statistical evaluation of the significance of magnitudes.

## CAT Cohort: Identification of MHC Class I-restricted epitopes

### Protected CAT participant v12 (HLA A*02:02 (A02)/A*32:01 (A01), B*35:01 (B07)/ B*44:03 (B44): (Tables 3 and S4-S6)

*Synthesized and confirmed predicted minimal epitope within 15mer component of AMA1 subpool*. **AMA1 Ap8 subpool**: Positive IFN-γ responses to 15mer peptide A97 within Ap8 subpool (Table 3) contained predicted HLA A*32:01 (A01 ST)-restricted epitope **KSHGKGYNW** that recalled IFN-γ responses. **KSHGKGYNW** is promiscuous as it recalled B*58:01 (B58 ST)-restricted responses from protected CA v90 (Table 1). Protected CAT participant v12 also

**Table 3. Cohort CAT: FluoroSpot IFN-γ and GzB responses for protected participant v12 (HLA A*02:02 (A02)/A*32:01 (A01), B*35:01 (B07)/B*44:03 (B44) to AMA1 Ap8 15mer peptide and synthesized predicted epitope.**

| Pool/ 15mer | 15mer Sequence | IFN-γ sfc/m | GzB sfc/m | HLA restriction | 15mer peptide | Epitope | IFN-γ sfc/m | GzB sfc/m |
|---|---|---|---|---|---|---|---|---|
| **Ap8** | | **158** | 0 | | | | | |
| A97 | FKADRY(**KSHGKGYNW**) | **188** | 15 | **A*32:01** | **A97** | RYKSHGKGY | 15 | 28 |
| | | | | | | **DRYKSHGKGY** | **40** | 70 |
| | | | | | | **YKSHGKGYNW** | **168** | 135 |
| | | | | | | FKADRYKSH | 25 | 65 |
| | | | | | | KADRYKSHGK | 15 | 75 |
| | | | | | | **KSHGKGYNW** | **88** | 50 |
| | | | | | | **DRYKSHGK** | **113** | 115 |
| | | | | | | **ADRYKSHGK** | 0 | 38 |

PBMCs were collected from participant v12 post-ChAd63/pre-CHMI. All 15mer peptides within Ap8 were tested in FluoroSpot assays. Positive activities are shown in bold. Predicted epitope within the positive 15mer is shown in bold with parenthesis and underlined. Each predicted epitope shown was synthesized and tested in FluoroSpot assays.

recognized four additional HLA A01 ST-restricted epitopes that overlapped with portions of the **KSHGKGYNW** sequence (Tables 3 and S4).

*Epitopes predicted within active CSP and TRAP 15mer peptides not tested.* Additional vaccine antigen-specific responses against minimal epitopes within active 15mer peptides components of immunodominant CSP and TRAP subpools could have been involved in the overall sterile protection obtained in CAT participant v12. However, due to limitation of PBMCs, such predicted epitopes were not tested to confirm association with protection.

**CSP Cp5 subpool**: (S5 Table): IFN-γ responses against CSP 15mer peptide C45 from Cp5 subpool contains predicted HLA B*35:01 (B07 ST)-restricted epitope **EPSDKHIKEY** that overlaps with the HLA A*01:01 (A01 ST)-restricted epitope **PSDKHIKEY**. These epitopes were not tested for confirmation.

**TRAP TD1 subpool:** (S6 Table): IFN-γ, IFN-γ and GzB, or GzB-only responses to 19 of the 25 TD1 15mer peptides (designated as SS-1 to SS-25 in S6 Table) were active and contained epitopes predicted to bind HLA A01, HLA A02, HLA B07 and HLA B44-restricted epitopes. None of these epitopes have been synthesized and tested to confirm activity.

**TRAP TD2 subpool**: IFN-γ, IFN-γ and GzB, or GzB-only responses to 10 of the 25 TD2 15mer peptides (designated as SS-25 to SS-50 in S6 Table) were active and contained epitopes predicted to bind HLA A01, HLA A02, HLA B07 and HLA B44-restricted epitopes. None of these epitopes have been synthesized and tested to confirm activity.

**Protected CAT participant v24 (HLA A*02:01 (A02)/A*30:01 (A01A03), B*35:01 B07/ B*42:02) (B07): (Tables 4, S7 and S8)**

*Synthesized and confirmed predicted minimal epitope within 15mer component of AMA1 subpool.* **AMA1 Ap1 subpool**: IFN-γ responses to 15mer peptides A11 and A12 within Ap1 subpool that contained predicted HLA B*42:01 (B07 ST)-restricted epitope **HPKEYEYPL** and recalled IFN-γ and GzB responses (Table 4).

**AMA1 Ap3 subpool**: IFN-γ responses to 15mer peptide A36 within Ap3 subpool contained predicted HLA B*35:01(B07 ST)-restricted epitope **LPSGKCPVF** that recalled IFN-γ and GzB responses. However, another predicted HLA B*35:01(B07 ST)-restricted epitope **IIIENSNTTF** within 15mer peptide A39 from Ap3 subpool did not recall responses from same subject v24.

**Table 4. Cohort CAT: FluoroSpot IFN-γ and GzB responses for protected participant v24 (HLA A*02:01 (A02)/A*30:01 (A01A03), B*35:01 (B07)/B*42:02 (B07) to AMA1 Ap1, Ap3 and Ap9 peptide pools, 15mer peptides, and synthesized predicted epitopes.**

| Pool/ 15mer | 15mer Sequence | IFN-γ sfc/m | GzB sfc/m | HLA restriction | 15mer peptide | Epitope | IFN-γ sfc/m | GzB sfc/m |
|---|---|---|---|---|---|---|---|---|
| **Ap1** | | **298** | 0 | | | | | |
| A11 | INEHRE(**HPKEYEYPL**) | **160** | 19 | **B*42:01** | **A11/A12** | **HPKEYEYPL** | **293** | **40** |
| A12 | RE(**HPKEYEYPL**)HQEH | **130** | 0 | **B*42:01** | | | | |
| **Ap3** | | 38 | 0 | | | | | |
| A36 | QYR(**LPSGKCPVF**)GKG | **115** | **0** | **B*35:01** | **A36** | **LPSGKCPVF** | **310** | **77** |
| A39 | GKG(**IIIENSNTTF**)LT | **100** | 71 | **B*35:01** | **A39** | IIENSNTTF | 10 | 17 |
| | | | | | | IIIENSNTTF | 3 | 3 |
| **Ap9** | | **418** | 0 | | | | | |
| A108 | TALS(**HPIEVENNF**)PC | **515** | 0 | **B*35:01** | **A108/A109** | **HPIEVENNF** | **687** | **140** |
| A109 | (**HPIEVENNF**)PCSLYK | **238** | 0 | **B*35:01** | | | | |

PBMCs were collected from participant v24 post-ChAd63/pre-CHMI. All 15mer peptides within Ap1, Ap3 and Ap9 were tested in FluoroSpot assays. Positive activities are shown in bold. Predicted epitopes within positive 15mers are shown in bold with parenthesis and underlined. Each predicted epitope shown was synthesized and tested in FluoroSpot assays.

**AMA1 Ap9 subpool**: IFN-γ responses to overlapping 15mer peptides A108 and A109 contained predicted HLA B*35:01(B07 ST)-restricted epitope **HPIEVENNF** that recalled IFN-γ and GzB responses.

*Predicted minimal epitopes within antigen-specific 15mer subpools not tested for confirmation*. **AMA1 Ap1 subpool**: (S7 Table): IFN-γ responses to overlapping 15mer peptides A7 and A8 that contain predicted HLA B*35:01 (B07 ST)-restricted epitope **HPYQNSDVY**. These epitopes were not tested for confirmation.

**TRAP TD1 subpool** (designated as SS-1 to SS-25 in S8 Table): Both protected v12 and protected v24 participants express HLA A02 and HLA B07 supertypes with different allelic types and they both recognized some, predicted epitopes, but not all. v24 IFN-γ responses only recognized SS-23, which contains a variable HLA B07-restricted epitope that was also recognized by protected v12 by GzB responses. However, there were no v24 responses to the remaining 18 TD1 15mers that were recognized by v12 (S8 Table).

**TRAP TD2 subpool (**designated as SS-26 to SS-50 in S8 Table): v24 IFN-γ responses recognized three of five TD2 15mers that were also recognized by v12. V24 also induced GzB-only responses against two TD2 15mers which contain predicted HLA B07 and HLA A01A03 epitopes that were not recognized by v12.

**TRAP TD5 subpool** (designated as SS-98 to SS-122 in S8 Table: IFN-γ responses to TD5 15mer SS-104 that contains predicted HLA A01A03 ST-restricted epitope **RYIPYSPLS**.

**Non-protected participant v33 (HLA A*23:01 (A24)/A*33:03 (A03), B*51:01 (B07)/ B*58:01 (B58): (Tables 5, S9 and S10)**

*Synthesized and confirmed predicted minimal epitope within 15mer component of AMA1 subpool*. **AMA1 Ap8 subpool**: Positive IFN-γ and GzB responses to two overlapping 15mer peptides A97 and A98 within Ap8 subpool that contained predicted HLA B*58:01 (HLA B58 ST)-restricted epitope **KSHGKGYNW** recalled IFN-γ and GzB responses (Table 5). However, the second overlapping 15mer A98 that contained a predicted HLA B*35:01 (HLA B07 ST)-restricted epitope **HGKGYNWGNY,** that did not recall responses, confirming that the dominant minimal epitope is **KSHGKGYNW** (Table 5).

**AMA1 Ap10 subpool**: IFN-γ responses Ap10 peptide A126 that contains HLA A*33:03 (A03 ST)-restricted epitope **RFFVCKCVER** recalled IFN-γ and GzB responses (Table 5).

**Table 5. Cohort CAT: FluoroSpot IFN-γ and GzB responses for non-protected participant v33 (HLA A*23:01 (A24)/A*33:03 (A03), B*51:01 (B07)/B*58:01 (B58) to AMA1 Ap4, Ap8 and Ap10 peptide pools, 15mer peptides, and synthesized predicted epitopes.**

| Pool/ 15mer | 15mer Sequence | IFN-γ sfc/m | GzB sfc/m | HLA restriction | 15mer Peptide | Epitope | IFN-γ sfc/m | GzB sfc/m |
|---|---|---|---|---|---|---|---|---|
| **Ap4** | | **110** | 6 | | | | | |
| A50 | E(**MRHFYKDNKY**)VKNL | **145** | **168** | A*33:03 | A50 | MRHFYKDNKY | 0 | 0 |
| **Ap8** | | **973** | **170** | | | | | |
| A97 | FKADRY(**KSHGKGYNW**) | **1123** | **130** | B*58:01 | A97 | **YKSHGKGYNW** | 480 | 170 |
| | | | | | | **KSHGKGYNW** | 975 | 303 |
| | | | | | | **DRYKSHGK** | 8 | 35 |
| A98 | RYKS(**HGKGYNWGNY**)N | **395** | 4 | B*35:01 | A98 | HGKGYNWGNY | 0 | 0 |
| **Ap10** | | **68** | 0 | | | | | |
| A126 | TC(**RFFVCKCVER**)RAE | 33 | 0 | A*33:03 | A126 | **RFFVCKCVER** | 160 | 113 |

PBMCs were collected from participant v33 post-ChAd63/pre-CHMI. All 15mer peptides within AMA1 subpools that include Ap4, Ap8 and Ap10 were tested in FluoroSpot assays. Positive IFN-γ and GzB activities of individual 15mer peptides within the subpools are shown in bold. Predicted epitopes within positive 15mers are also shown in bold with parenthesis and underlined. Each predicted epitope shown was synthesized and tested in FluoroSpot assays with the parent 15mer and AMA1 peptide pool.

**Table 6. Cohort CAT: FluoroSpot IFN-γ responses for non-protected participant v35 (HLA A\*02:01 (A02)/A\*02:06 (A02), B\*39:05 (B27)/B\*58:01 (B58) to AMA1 Ap8 peptide pool, 15mer component peptides, and synthesized predicted epitopes within active 15mers.**

| Pool/ 15mer | 15mer Sequence | IFN-γ sfc/m | GzB sfc/m | HLA restriction | 15mer Peptide | Epitope | IFN-γ sfc/m | GzB sfc/m |
|---|---|---|---|---|---|---|---|---|
| **Ap8** | | **325** | 25 | | | | | |
| A97 | FKADR(**YKSHGKGYNW**) | **470** | **53** | **B\*58:01** | **A97** | **YKSHGKGYNW** | **135** | **85** |
| | | | | | | **KSHGKGYNW** | **393** | 30 |

PBMCs were collected from participant v35 post-ChAd63/pre-CHMI. All 15mer peptides within Ap8 were tested in FluoroSpot assays. Positive activities are shown in bold. Predicted epitopes within positive 15mers are shown in bold with parenthesis and underlined. The predicted epitope shown was synthesized and tested in FluoroSpot assays.

*Synthesized predicted minimal epitope within strongly active 15mer component of AMA1 subpool that was not confirmed upon testing.* **AMA1 Ap4 subpool**: IFN-γ and GzB responses to Ap4-A50 contains predicted HLA A\*33:03 (A03 ST)-restricted epitope **MRHFYKDNKY** that did not recall responses (Table 5).

In addition to AMA1 responses, non-protected participant v33 also induced responses against some TRAP 15mers predicted to contain HLA B07- and B58-restricted epitopes but they were not tested (S10 Table).

**Non-protected participant v35 (HLA A\*02:01 (A02)/A\*02:06 (A02), B\*39:05 (B27)/ B\*58:01 (B58): (Tables 6, S11 and S12)**

*Synthesized and confirmed predicted minimal epitope within immunodominant 15mer component of AMA1 subpool.* **AMA1 Ap8 subpool**: Positive IFN-γ and GzB responses to 15mer peptide A97 that contained predicted HLA B\*58:01 (B58 ST)-restricted epitope **KSHGKGYNW** that recalled IFN-γ responses, and the longer overlapping synthesized epitope **YKSHGKGYNW** recalled IFN-γ and GzB responses (Table 6).

In addition to AMA1 responses, non-protected participant v35 induced additional responses against some TRAP 15mer peptides predicted to contain HLA B27, and B58, and A02-restricted epitopes but those epitopes were not synthesized and tested (S12 Table). Interestingly, TRAP 15mer peptides were predicted to contain four HLA A02-restricted epitopes that were recognized by protected v24 (S8 Table) were not recognized by v35 (S12 Table).

## Interpretation of responses of CAT Cohort protected v12, v24 and non-protected v33 and v35

The results of our antigen-specific subpool responses showed that immunodominant responses were induced against limited number of AMA1 subpools as reported previously [10] that include Ap1, Ap8, and Ap10 (Fig 1). Due to limitation of cells, we therefore concentrated on identifying protection associated epitopes within the immunodominant subpools, specifically, on AMA1 subpools Ap1, Ap8, and Ap10.

- Possible association of tested class-I restricted epitope responses with protection:

○ Protection of v12 (HLA A02/A01, B07/B44) is associated with IFN-γ responses to the **KSHGKGYNW**, by promiscuous binding through HLA A01 ST. We previously reported that response to **KSHGKGYNW** was associated with protection of an HLA B58 participant in the DNA/HuAd5 trial [10].

○ Protection of v24 is associated with IFN-γ and GzB responses to the AMA1 HLA B\*42:01 (B07 ST)-restricted epitope **HPKEYEYPL**, to AMA1 HLA B\*35:01 (HLA B07 ST)-

restricted epitope **HPIEVENNF**, and to AMA1 B*35:01 (B07 ST)-restricted epitope **LPSGKCPVF**.

- Co-expression of additional HLA alleles may influence whether responses are associated with protection:

○ Two protected participants in this trial, v90 (CA cohort) and v12 (CAT cohort), and two protected participants in the DNA/HuAd5 trial [18] each express HLA B58 and recognize the HLA B58-restricted **KSHGKGYNW**, except v12, which recognizes **KSHGKGYNW** by promiscuous binding through HLAA01. Each of these four protected participants also expressed HLA B44.

○ In contrast, two non-protected participants (v33 and v35) also expressed HLA B58, recognized epitope **KSHGKGYNW**, but did not express HLA B44. Significance of this finding is unknown, and the sample size studied is too small to draw a conclusion from this observation.

○ A third participant in the DNA/HuAd5 trial recognized the HLA A03-restricted epitope **STCRFFVCK** and expressed HLA B44, whereas non-protected v32 in this trial also recognized the HLA A03-restricted epitope **STCRFFVCK** but did not express HLA B44. Furthermore, the protected DNA/HuAd5 trial participant expressed the HLA A*11:01 allele, while CAT v32 in this trial expressed A*03:03 allele.

○ In a separate trial, one participant immunized only with HuAd5 containing CA had a significant delay to infection and recognized the HLA B58-restricted **KSHGKGYNW** but also lacked HLA B44 [6].

○ We suggest that interactions between HLA B44 with HLA B58 or HLA A01 may influence protective responses.

○ Recognition of TRAP TD1 and TD3 peptides was also influenced by co-expression of different HLA alleles: v24 and v33 shared common HLA alleles with v35 but different co-expressed HLA alleles.

- The breadth of epitope responses may influence protection:

○ IFN-γ and GzB responses in protected participants v12 and v24 to TRAP were broader, recognizing more 15mers in TD1/TD2 and likely more HLA-restricted epitopes compared to non-protected participants v33 and v35. Interestingly, it appears that strong responses to TD3 were not associated with protection since the two subjects (v33 and v35) with the highest TD3 subpool responses (Fig 1) were not protected.

- Conclusion for TRAP:

○ Although we could not test individual TRAP epitopes, strong responses to 15mer peptides suggest that Trap epitopes may have played a role in the higher VE of the CAT cohort compared to the CA cohort.

## Summary of confirmed epitopes

Overall, 10 epitopes predicted by the NetMHCpan-4.0 algorithm were evaluated in FluoroSpot assays, and seven epitopes recalled responses, whereas three epitopes did not recall responses. We and others have previously suggested that epitope validation should require evaluation in immunological assays to verify prediction [16,21,24].

**Table 7. CA and CAT Cohorts: Summary of seven confirmed HLA-restricted epitopes compared to previous clinical trials.**

| Cohort | Participant | Status | HLA | Antigen | Epitope | IFN-γ/GzB | Restriction | Promiscuity[2] |
|---|---|---|---|---|---|---|---|---|
| DNA/ChAd63 CA | **v90** | P | **A03/A03**, B44[5]/**B58** | AMA1 | **KSHGKGYNW** | IFN-γ | **B58** | A01 |
| | | | | AMA1 | **RFFVCKCVER** | IFN-γ/GzB | **A03** | |
| | **v32** | NP | **A03/A03**, B07/B07 | AMA1 | **STCRFFVCK** | IFN-γ/GzB | **A03** | |
| | | | | AMA1 | **SAFLPTGAFK** | IFN-γ | **A03** | B07, B27, B62 |
| DNA/ChAd63 CAT | v12 | P | A02/**A01**, B07/B44[5] | AMA1 | **KSHGKGYNW** | IFN-γ | **A01** | B58 |
| | v24 | P | A02/A01 A03, **B07**/B42[4] | AMA1 | **HPKEYEYPL** | IFN-γ/GzB | **B07** | B08, B27, B44 |
| | | | | AMA1 | **HPIEVENNF** | IFN-γ/GzB | **B07** | |
| | | | | AMA1 | **LPSGKCPVF** | IFN-γ/GzB | **B07** | |
| | **v33** | NP | A24/**A03**, B07/**B58** | AMA1 | **KSHGKGYNW** | IFN-γ/GzB | **B58** | A01 |
| | | | | AMA1 | **RFFVCKCVER** | IFN-γ/GzB | **A03** | |
| | **v35** | NP | A02/A02, B27/**B58** | AMA1 | **KSHGKGYNW** | IFN-γ | **B58** | A01 |
| DNA/ HuAd5 CA | **v10** | P | A01/A01, B44[5]/**B58** | AMA1 | **KSHGKGYNW** | IFN-γ[1] | **B58** | A01 |
| | **v11** | P | **A03/A03**, B44[5]/B07 | AMA1 | **STCRFFVCK** | IFN-γ[1] | **A03** | |
| | V18 | P | A02/A02, B44[5]/**B58** | AMA1 | **KSHGKGYNW** | IFN-γ[1] | **B58** | A01 |
| HuAd5 CA | V194 | **NP[3]** | A01/A03, B27/**B58** | AMA1 | **KSHGKGYNW** | IFN-γ[1] | **B58** | A01 |

All confirmed epitopes in six DNA/ChAd63 CA and CAT participants (this trial) are compared with three protected participants in the DNA/HuAd5 CA trial [10], and a participant in the HuAd5 CA trial that had a significant delay to parasitemia [5]. HLA alleles of confirmed epitopes that match those of that subject are in bold. HLA B44 is included to show association with protection.

[1]Only IFN-γ responses and not GzB responses were measured.

[2]Promiscuous alleles are shown, a blank indicates that the epitope is not promiscuous.

[3]Significant delay to infection, but not sterile protection.

[4]B42 is unclassified as a ST.

[5]B44 is associated with protection.

All seven HLA-restricted epitopes that were confirmed by FluoroSpot assays are summarized in Table 7, and all were within AMA1. Five of the seven confirmed epitopes were associated with both IFN-γ and GzB responses (**RFFVCKCVER, STCRFFVCK, HPIEVENNF, HPKEYEYPL,** and **LPSGKCPVF**), one epitope was associated with IFN-γ and/or GzB responses depending on the participants' HLA restriction (**KSHGKGYNW**), and one epitope (**SAFLPTGAFK**) was associated only with IFN-γ responses.

Given the small sample size it is not possible to definitively determine if these epitopes are associated or not associated with protection. However, **HPKEYEYPL**, **LPSGKCPVF**, and **HPIEVENNF**, recalled responses from protected participants, and **SAFLPTGAFK** and **STCRFFVCK** recalled responses from non-protected participants. Two epitopes **KSHGKGYNW** and **RFFVCKCVER** recalled responses from protected and non-protected participants but were associated with protection when that participant also expressed HLA B44 and were associated with non-protection when HLA B44 was absent. We suggest that HLA B44, or other unidentified HLA alleles, may also determine protection.

## Summary of predicted epitopes that were not tested

IFN-γ or GzB responses identified two epitopes in CSP and two epitopes in AMA1 that were not synthesized and tested. When all IFN-γ and GzB responses to individual TRAP 15mers were compared (S13 Table), a total of 35 predicted but not confirmed epitopes were identified, predominantly in TD1, with fewer in TD2, TD3 and TD5, as found in other studies [25]. GzB-only responses recognized 19 of these 35 predicted TRAP epitopes. None of these predicted epitopes were synthesized and therefore cannot be confirmed.

## Discussion

Sterile protection to malaria is mediated, at least in part, by CD8+ T cells that recognize MHC Class I-restricted malaria antigen epitopes on the surface of infected hepatocytes and secrete IFN-γ, GzB or other lytic agents [2–5]. These MHC Class I-specific responses are probably generated by cross-presentation of the vaccine antigens [26–28], as well as hepatocellular antigen presentation [29], neither of which are fully understood in malaria. Our rationale for identifying CSP, AMA1 and TRAP epitopes is to construct a multi-epitope malaria vaccine designed to induce long-lasting protective CD8+ T cell responses that may be superior to current antibody-based vaccines, or to supplement these vaccines.

Using the mouse malaria model *P. yoelii*, we have previously demonstrated that the H2$^d$-restricted *P. yoelii* epitope **SYVPSAEQI** induced cytotoxic T cell responses in mice and 40% protection to sporozoite challenge [30]. We concluded that this epitope was protective as it experimentally induced protective immune responses. In our previous DNA/HuAd5 CA trial, the AMA1 vaccine antigen induced protection in four participants, three of whom developed ELISpot IFN-γ and CD8+ T cell responses to a single AMA1 HLA B58-restricted epitope **KSHGKGYNW** (two protected subjects) and another AMA1 HLA A03-restricted epitope (one protected subject) **STCRFFVCK** epitope [18]. However, unlike the earlier experiments in mice, we have not evaluated immunogenicity of formulated individual AMA1 epitopes in human clinical experiments, so we have not demonstrated formal confirmation that these epitopes themselves will be immunogenic and induce protection. In these clinical experiments we can only demonstrate that ELISpot IFN-γ responses individual antigen-specific epitopes developed after immunization with gene-based vaccines containing full whole antigens in protected but non-protected participants. Therefore, in human experiments, we are careful to restrict our conclusions that these epitopes are associated with protection, but we cannot definitively conclude they are protective epitopes in the absence of direct human immunization and resistance to sporozoite challenge experiments.

A defining characteristic of three of four protected participants in the DNA/HuAd5 trial was their focused response (immunodominance) to single CSP or AMA1 peptide subpools, allowing identification of protection-associated epitopes in AMA1 protein. We were unable to conduct a similar analysis to identify protection associated epitopes in CSP due to limitation of cells[10]. In this trial we also found that the single protected participant in the CA cohort, v90, had focused immunodominant IFN-γ and GzB responses to a sequence in AMA1 protein. In the current CAT cohort, two of five protected participants had focused immunodominant IFN-γ and/or GzB responses to CSP (v12:Cp5 subpool), AMA1(v12:Ap8; v24:Ap1 and TRAP (v12 & v24:TT2/TD2).

It is unclear whether these differences reflect the small numbers of protected participants, replacement HuAd5 with ChAd63, the addition of TRAP, or other factors. An earlier study using ChAd63/MVA ME-TRAP identified HLA A03-restricted responses directed to TRAP T99/96 peptides containing a putative epitope [31] adjacent to the B44-restricted TRAP epitope **RENANQLVV** identified here (S6 Table). It is possible that these additional TRAP epitopes may contribute to the superior efficacy of the CAT vaccine compared to the CA vaccine. However, some predicted TRAP epitopes are variable and allelic polymorphism may influence the induction of T cell responses against MHC Class I-restricted malaria epitopes that are critical for protective immunity [32,33].

Immunodominance may occur with co-expression of another MHC HLA Class I molecule [34]. The two protected participants in this trial and three protected participants in previous DNA/HuAd5 trial co-expressed HLA B44. We do not have conclusive evidence that HLA B44 may play a role in HLA-restricted protected responses against malaria. Prediction models

suggest that AMA1 contains numerous HLA B44 epitopes, including predicted high binding epitopes that are highly conserved and may therefore be important for malaria vaccine design [33]. Further investigation into the fine specificity of HLA B44 expression and protection is warranted. This is supported by the apparent linkage disequilibrium of HLA B58 and HLA B44 in association with protection [35,36] and may also reflect evolution of HLA B44 and HLA B58 from a single ancestor, whereas HLA B07 and HLA B27 evolved from different multiple origins [37].

We also found that fine HLA allele specificity was a crucial determinant of protective responses. Protective responses to **STCRFFVCK** were associated with HLA A*11:01 responses in the DNA/HuAd trial [18], but not HLA A*03:01 even though both are members of the A03 ST. Similarly, protective responses to **RFFVCKCVER** were associated with HLA A*33:03 responses and not HLA A*03:01, again both members of the HLA A03 ST. This differs from epitope promiscuity, where for example epitope **KSHGKGYNW** was recognized by HLA B58 and HLA A01-restricted responses. In fact, three of the seven confirmed AMA1 epitopes were promiscuous.

We found that IFN-γ and GzB responses, IFN-γ-only responses, and GzB-only responses occurred with CSP, AMA1 and TRAP peptide subpools, and with individual epitopes. GzB-only responses to peptide subpools were rare and only two participants in the CA cohort made unique GzB-only responses to CSP subpools. When subpool 15mers were individually tested, we found that GzB-specific responses were only made to one AMA1 15mer (Ap8-A103). One shorter synthesized epitope **(DRYKSHGK)** induced GzB-only responses from AMA1 and none from CSP. However, GzB-only responses were made to 19 of 36 TRAP 15mer peptides. Since only a subset of participants were tested it is likely that GzB responses may be more frequent and responses to peptide subpools do not fully represent responses to their component 15mers. We suggest that further studies are needed to clarify the significance of GzB responses after malaria vaccine immunization or natural transmission.

We have previously shown that CD8+ T cell depletion studies using ELISpot abrogate most responses, indicating that CD8+ T cells constitute the predominant response induced by these DNA/Ad vaccines [18]. Cytotoxic CD8+ T cells kill target cells, in this case malaria infected hepatocytes, by secreted IFN-γ or by direct cell-cell contact releasing GzB and other lytic particles [38,39]. Vaccine studies, for example using influenza, have shown a significant correlation between IFN-γ and GzB [40]. It remains unclear whether CD8+ T cells produce IFN-γ and GzB together or separately [41–44], but the differential association of IFN-γ or GzB with protection suggests that CD8+ T cells produce each separately, in agreement with studies in non-human primates and SIV [42]. Cell surface markers such as CD29 could distinguish CD8+ T cells with high levels of such cytotoxic molecules [43].

Overall, we found that confirmation of the possible role of MHC Class I epitopes required functional assays, such as FluoroSpot, in addition to using predictive algorithms, corroborating our [21,45,46] and other earlier findings. Other epitopes have been predicted *in silico* but remain to be functionally verified [47–53]. This may be critical for design of a Pf epitope ensemble vaccine [53–55]. This is also important to better understand immune evasion strategies where single amino acid changes in T cell epitopes lead to loss of binding to the MHC complex, lack of cross-reactivity and immune interference [21,56].

## Conclusions

The DNA/ChAd63 vaccine induced superior protection in humans using the three-antigen combination CSP, AMA1 and TRAP compared to the earlier DNA prime-recombinant viral boost malaria vaccine containing only CSP and AMA1. We have confirmed the importance of

two previously identified HLA B58 and HLA A03-restricted AMA1 epitopes and extended these findings to include additional confirmed AMA1 epitopes and predicted CSP and TRAP epitopes. It is likely that protection-associated TRAP epitopes enhanced vaccine efficacy of the three-antigen combination and additional assays such as flow/intracellular staining of T cell subsets may further refine these observations.

## Limitations

The major limitation to this study is the availability of frozen PBMCs from vaccine-immunized protected and non-protected participants. This resulted in a more limited analysis of antigen-specific epitope-specific responses than we would have preferred for this trial. Another limitation is the exclusive use of the FluoroSpot IFN-γ and GzB assays; other assays including flow/intracellular staining that requires more cells used to identify T cell subsets may be required to further investigate these participants.

## Supporting information

**S1 Table. Cohort CA and Cohort CAT participants: HLA alleles and supertypes.**
(DOCX)

**S2 Table. Cohort CA: FluoroSpot IFN-γ and GzB responses for protected participant v90 (HLA A03/A03, B44/B58) to AMA1 Ap8, and Ap10 peptide sub pools, single 15mer peptides, and synthesized predicted minimal epitopes.**
(DOCX)

**S3 Table. Cohort CA: FluoroSpot IFN-γ and GzB responses for non-protected participant v32 (HLA A03/A03, B07/B07) to AMA1 Ap8 and Ap10 subpools, 15mer peptides, and synthesized predicted epitopes.**
(DOCX)

**S4 Table. Cohort CAT: FluoroSpot IFN-γ and GzB responses for protected participant v12 (HLA A02/A01, B07/B44) to AMA1 subpool Ap8, 15mer peptides and synthesized predicted epitopes.**
(DOCX)

**S5 Table. Cohort CAT: FluoroSpot IFN-γ and GzB responses for protected participant v12 (HLA A02/A01, B07/B44) to CSP subpool Cp5 and 15mer peptides.**
(DOCX)

**S6 Table. Cohort CAT: FluoroSpot IFN-γ and GzB responses for protected participant v12 (HLA A02/A01, B07/B44) to 3D7 TRAP TD1 and TD2 sub pools and positive 15mer peptides containing predicted epitopes.**
(DOCX)

**S7 Table. Cohort CAT: FluoroSpot IFN-γ and GzB responses for protected participant v24 (HLA A02/A01A03, B07/B42 [unclassified]) to AMA1 Ap1, Ap3 and Ap9 subpools, 15mer peptides, and synthesized predicted epitopes.**
(DOCX)

**S8 Table. Cohort CAT: FluoroSpot IFN-γ and GzB responses for protected participant v24 (HLA A02/A01A03, B07/B42 [unclassified]) to 3D7 TRAP TD1, TD2 and TD5 subpools and 15mer peptides containing predicted epitopes.**
(DOCX)

**S9 Table. Cohort CAT: FluoroSpot IFN-γ and GzB responses for non-protected participant v33 (HLA A24/A03, B07/B58) to AMA1 Ap4, Ap 8 and Ap 10 subpools,15mer peptides, and synthesized predicted epitopes.**
(DOCX)

**S10 Table. Cohort CAT: FluoroSpot IFN-γ and GzB responses for non-protected participant v33 (HLA A24/A03, B07/B58) to 3D7 TRAP TD3 peptide pool and 15mer peptides containing predicted epitopes.**
(DOCX)

**S11 Table. Cohort CAT: FluoroSpot responses for non-protected participant v35 HLA A02/A02, B27/B58 to AMA1 Ap8 peptide pool, 15mer peptides, and synthesized predicted epitopes.**
(DOCX)

**S12 Table. Cohort CAT: FluoroSpot responses for non-protected participant v35 HLA A02/A02, B27/B58 to TRAP TD1 and TD3 peptide pools, 15mer peptides, and predicted epitopes.**
(DOCX)

**S13 Table. Summary of predicted but not confirmed CSP, AMA1 and TRAP MHC Class I-restricted epitopes.**
(DOCX)

## Acknowledgments

MS, JEE, KAE, EV and CAD are employees of the U.S. Government, and this work was prepared as part of their official duties. Title 17, U.S.C., §105 provides that copyright protection under this title is not available for any work of the U.S. Government. Title 17, U.S.C., §101 defines U.S. Government work as work prepared by a military Service member or employee of the U.S. Government as part of that person's official duties. The views expressed in this article are those of the authors and do not necessarily reflect the official policy or position of the Henry M. Jackson Foundation for the Advancement of Military Medicine, Inc., Department of the Navy, the Department of Defense, nor the U.S. Government. All authors have read and approved the final version of the manuscript. Author Judith Epstein is supported by the Intramural Research Program of the National Institute of Allergy and Infectious Diseases, National Institutes of Health.

## Author Contributions

**Conceptualization:** Noelle Patterson, Lorraine Soisson, Martha Sedegah.

**Formal analysis:** Bjoern Peters, Martha Sedegah.

**Funding acquisition:** Santina Maiolatesi, Lorraine Soisson, Eileen Villasante.

**Investigation:** Harini Ganeshan, Jun Huang, Maria Belmonte, Arnel Belmonte, Sandra Inoue, Rachel Velasco, Santina Maiolatesi, Keith Limbach, Noelle Patterson, Marvin J. Sklar, Judith E. Epstein, Christopher A. Duplessis, Martha Sedegah.

**Methodology:** Santina Maiolatesi.

**Resources:** Eileen Villasante.

**Supervision:** Keith Limbach, Marvin J. Sklar, Martha Sedegah.

**Visualization:** Michael R. Hollingdale.

**Writing – original draft:** Michael R. Hollingdale.

**Writing – review & editing:** Judith E. Epstein, Kimberly A. Edgel, Bjoern Peters, Michael R. Hollingdale, Eileen Villasante, Christopher A. Duplessis, Martha Sedegah.

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
