## [Decision Letter · Decision Letter 0]

9 Jul 2024

PONE-D-24-16201The DNA prime/chimpanzee adenovirus (ChAd63) boost vaccine confers sterile immunity associated with CSP, AMA1 and TRAP MHC Class I-restricted epitopesPLOS ONE

Dear Dr. Sedegah,

Thank you for submitting your manuscript to PLOS ONE. After careful consideration, we feel that it has merit but does not fully meet PLOS ONE’s publication criteria as it currently stands. Therefore, we invite you to submit a revised version of the manuscript that addresses the points raised during the review process.

Please prepare the Authors' Reply based on the 'comments from Reviewer 1 and Reviewer 3' and the 'revised comments from Reviewer 2 shown below'.

We look forward to receiving your revised manuscript.

Kind regards,

Takafumi Tsuboi

Academic Editor

PLOS ONE

Journal Requirements:

3. Please include your tables as part of your main manuscript and remove the individual files. Please note that supplementary tables (should remain/ be uploaded) as separate "supporting information" files.

Additional Editor Comments:

As Reviewer 2 had a problem with the initial Review downloading the manuscript file, Reviewer 2 re-submitted the Reviewer's comments after the data was available.

Therefore, please consider the following comments from Reviewer 2 in this window to prepare for the Author reply.

Revised comments from the Reviewer 2:

In this manuscript titled “The DNA prime/chimpanzee adenovirus (ChAd63) boost vaccine confers sterile immunity associated with CSP, AMA1 and TRAP MHC Class I-restricted epitopes”, the authors aimed to determine the contribution of MHC class I epitopes derived from CSP, AMA1, and TRAP malaria antigens to vaccine efficacy (VE).

This manuscript contains a substantial amount of data using CHMI samples and seems informative in clarifying the mechanisms of malaria vaccines, especially the epitopes which are responsible to induce T cell response. However, first, a critical defect was found, so I would like to ask the authors to supply enough information for the review. Additionally, to justify their statement, I would like to ask the authors to explain more in detail. So, please refer to my questions and comments below.

1. Each experimental result on the manuscript is reasonable, and I agree with their confirmation of their epitope prediction. The authors could find some promising epitopes which can induce T cell IFN-g and GzB production with each donor sample. So, it means that the authors could show the utility of their prediction algorithms. But the authors should explain how they can justify the relationship between the immunogenicity of peptides and protection. (It is possible that other immune mechanisms like humoral immunity are responsible for the protection.) Similarly, even if CD8 T cell activating peptides are included in the TRAP sequence, its role in protection remains still unclear.

2. The authors predicted and confirmed MHC Class-I restricted epitopes which stimulate T cell and induce IFN-g and GzB production. But authors described “some non-protected individuals recognized HLA-matched protective minimal epitopes. (Line 53-54). How do authors think the reliability of the prediction?

3. Sample number is limited, and it cannot cover HLA various alleles. How do authors think the universality of your findings?

4. According to the 1-3, their title “The DNA prime/chimpanzee adenovirus (ChAd63) boost vaccine confers sterile immunity associated with CSP, AMA1 and TRAP MHC Class I-restricted epitopes” seems too strong.　

5. I could not find the legend for Figure 1. Therefore, it is difficult to provide a final evaluation of this manuscript with the information provided.

6. More detailed information of CHMI and definition of protected and non-protected should be provided in the methods part (even I could see this study following Reference 17).

Reviewers' comments:

Reviewer's Responses to Questions

**Comments to the Author**

1. Is the manuscript technically sound, and do the data support the conclusions?

Reviewer #1: Yes

Reviewer #2: Partly

Reviewer #3: No

2. Has the statistical analysis been performed appropriately and rigorously? 

Reviewer #1: No

Reviewer #2: I Don't Know

Reviewer #3: No

3. Have the authors made all data underlying the findings in their manuscript fully available?

Reviewer #1: Yes

Reviewer #2: Yes

Reviewer #3: No

4. Is the manuscript presented in an intelligible fashion and written in standard English?

Reviewer #1: Yes

Reviewer #2: Yes

Reviewer #3: No

5. Review Comments to the Author

Reviewer #1: The authors need an opening statement to orientate the readers to malaria and the challenges phased in control, hence the need for vaccines.

Line 163, Immunological samples: what is the justification for collection of PBMC on day 27?

Line 187, Fresh of cryopreserved PBMCs we expect some differences in results, how was this addressed, especially in the results presented.

The authors mention the use of Mann-Whitney U test in their methods, However, results/ discussion no mention is made in relation to the Mann-Whitney test results.

Discussion: pervious studies using huAd5 in malaria, eg Marvin J Saklar, were not included how does this overall compare to other manuscripts?

Reviewer #2: In this manuscript titled “The DNA prime/chimpanzee adenovirus (ChAd63) boost vaccine confers sterile immunity associated with CSP, AMA1 and TRAP MHC Class I-restricted epitopes”, the authors aimed to determine the contribution of MHC class I epitopes derived from CSP, AMA1, and TRAP malaria antigens to vaccine efficacy (VE).

This manuscript contains a substantial amount of data using CHMI samples and seems informative in clarifying the mechanisms of malaria vaccines. However, a critical defect interferes with an appropriate review. I would like to ask the authors to supply enough information for the review. Additionally, I strongly recommend that the authors reconsider how to summarize and explain their findings and narrative. Please refer to my questions and comments below to improve the manuscript.

1. I could not find Tables 1 to 7 and the legend for Figure 1. Therefore, it is difficult to provide a final evaluation of this manuscript with the information provided.

2. This manuscript has only one main Figure (I could see the indication for Tables 1 to 7 in the manuscript, but I could not find the Tables) and many Supplementary Tables. With long explanations for each Supplementary Tables, it was difficult to determine the priority. Therefore, the authors should select a few key Tables (which are essential for their conclusions) from the Supplementary Tables and make these the main Tables. The manuscript should then emphasize these Tables and their explanations more.

3. In the Background section, the authors stated, “We concluded that adding ME-TRAP to a two-antigen (CA) formulation increased VE. (Line 89-90)” and they concluded “We have confirmed the importance of two previously identified HLA B58 and HLA A03-restricted AMA1 epitopes and extended these findings to include additional confirmed AMA1 epitopes and predicted CSP and TRAP epitopes. It is likely that protection-associated TRAP epitopes enhanced vaccine efficacy of the three-antigen combination and additional assays such as flow/intracellular staining of T cell subsets may further refine these observations. (Line 533-537)”. Although, it is difficult to clarify enough, author should discuss more on the reason why CAT shows better VE than CA.

Reviewer #3: I find this manuscript problematic to read. On the one hand, it abounds in technical information that requires exhaustive knowledge of the subject. On the other hand, the structure of the text is not too appropriate. Several sections allude to the supplement tables, and it is unknown if they are part of the main text or the legend of the tables.

I view the statistical analysis as absent. The statistical analysis section in the manuscript has only one sentence. It establishes that the study has considered the Mann-Whitney test, but it no longer appears in any presentation or discussion of results. I can't quickly identify the comparisons made with the Mann-Whitney test. Besides, if multiple comparisons are needed (I think so), the family-wise error should be corrected appropriately.

Another concern is the sample size in each group. On the one hand, the sample sizes are small, and the authors emphasize many individual particularities in several subjects, so I wonder about the validity of the inferences. On the other hand, my view is that the paper does not state a clear relationship between the two experimental groups and the protected/unprotected subjects.

6. PLOS authors have the option to publish the peer review history of their article (what does this mean?). If published, this will include your full peer review and any attached files.

Reviewer #1: No

Reviewer #2: No

Reviewer #3: No

---

## [Author Response · Author response to Decision Letter 0]

23 Aug 2024

The rebuttal letter with responses to the reviewers is attached as file "Response to Reviewers.docx".

---

## [Decision Letter · Decision Letter 1]

10 Jan 2025

Human response to the DNA prime/chimpanzee adenovirus (ChAd63) boost vaccine identify CSP, AMA1 and TRAP MHC Class I-restricted epitopes

PONE-D-24-16201R1

Dear Dr. Sedegah,

We’re pleased to inform you that your manuscript has been judged scientifically suitable for publication and will be formally accepted for publication once it meets all outstanding technical requirements.

Kind regards,

Takafumi Tsuboi

Academic Editor

PLOS ONE

Additional Editor Comments (optional):

The authors successfully made significant revisions and improved the manuscript significantly, in accordance with all three Reviewers' comments. Therefore, I made a decision based on the final comments of Reviewer 2 and my own assessment.

Reviewers' comments:

Reviewer's Responses to Questions

**Comments to the Author**

1. If the authors have adequately addressed your comments raised in a previous round of review and you feel that this manuscript is now acceptable for publication, you may indicate that here to bypass the “Comments to the Author” section, enter your conflict of interest statement in the “Confidential to Editor” section, and submit your "Accept" recommendation.

Reviewer #2: All comments have been addressed

2. Is the manuscript technically sound, and do the data support the conclusions?

Reviewer #2: Partly

3. Has the statistical analysis been performed appropriately and rigorously? 

Reviewer #2: Yes

4. Have the authors made all data underlying the findings in their manuscript fully available?

Reviewer #2: Yes

5. Is the manuscript presented in an intelligible fashion and written in standard English?

Reviewer #2: Yes

6. Review Comments to the Author

Reviewer #2: (No Response)

7. PLOS authors have the option to publish the peer review history of their article (what does this mean?). If published, this will include your full peer review and any attached files.

Reviewer #2: No

---

## [Editor Report · Acceptance letter]

14 Jan 2025

PONE-D-24-16201R1 

PLOS ONE

Dear Dr. Sedegah, 

I'm pleased to inform you that your manuscript has been deemed suitable for publication in PLOS ONE. Congratulations! Your manuscript is now being handed over to our production team.

Kind regards, 

on behalf of

Prof. Takafumi Tsuboi 

Academic Editor

PLOS ONE